# Iterated Deep Q-Network: Efficient Learning of Bellman Iterations for Deep Reinforcement Learning

## Abstract

Value-based reinforcement learning (RL) methods strive to obtain accurate approximations of optimal action-value functions. Notoriously, these methods heavily rely on the application of the optimal Bellman operator, which needs to be approximated from samples. Most approaches consider only a single Bellman iteration, which limits their power. In this paper, we introduce Iterated Deep Q-Network (iDQN), a new DQN-based algorithm that incorporates several consecutive Bellman iterations into the training loss. iDQN leverages the online network of DQN to build a target for a second online network, which in turn serves as a target for a third online network, and so forth, thereby taking into account future Bellman iterations. While using the same number of gradient steps, iDQN allows for better learning of the Bellman iterations compared to DQN. We evaluate iDQN against relevant baselines on 54 Atari 2600 games to showcase its benefit in terms of approximation error and performance. iDQN outperforms its closest baselines, DQN and Random Ensemble Mixture, while being orthogonal to more advanced DQN-based approaches.

## 1 Introduction

Deep value-based Reinforcement Learning algorithms have achieved remarkable success in various fields, from nuclear physics (Degrave et al., 2022) to construction assembly tasks (Funk et al., 2022). These algorithms aim at learning a function as close as possible to the optimal action-value function, on which they can build a policy to solve the task at hand. To obtain an accurate estimate of the optimal action-value function, the optimal Bellman operator is used to guide the learning procedure in the space of $Q$-functions (Bertsekas, 2019) through successive iterations, starting from any $Q$-function to the optimal action-value function. In Reinforcement Learning, as opposed to Dynamic Programing, the reward function and system dynamics are not assumed to be known (Bertsekas, 2015). This forces us to approximate the optimal Bellman operator with an empirical Bellman operator. This problem has received a lot of attention from the community (Fellows et al. (2021), Van Hasselt et al. (2016)). On top of that, the use of function approximation results in the necessity of learning the projection of the empirical Bellman operator's iteration on the space of approximators. In this work, we focus on the projection step.

We propose a way to improve the accuracy of the learned projection by increasing the number of gradient steps and samples that each $Q$-function estimate has been trained on. This idea, implemented in the training loss function, uses the same total number of gradient steps and samples than the classical approaches. At a given timestep of the learning process, this new loss is composed of the consecutive temporal differences corresponding to the following Bellman iterations needed to be learned, as opposed to DQN (Mnih et al., 2015), where only one temporal difference related to the first projection step is considered. Each temporal difference is learned by a different neural network, making this method part of the DQN variants using multiple $Q$ estimates during learning. Those

consecutive temporal differences are computed in a telescopic manner, where the online network of the first temporal difference is used to build a target for the second temporal difference and so on. This loss implicitly incurs a hierarchical order between the $Q$ estimates by forcing each $Q$ estimate to be the projection of the Bellman iteration corresponding to the previous $Q$ estimate, hence the name *iterated Deep Q-Network* (iDQN). In the following, we start by reviewing algorithms built on top of DQN, highlighting their behavior in the space of $Q$-functions. We then introduce a new approach to Q-learning that emerges naturally from a graphical representation of DQN. In Section 5, we show the benefit of our method on the Arcade Learning Environment benchmark (Bellemare et al., 2013). Our approach outperforms DQN and Random Ensemble Mixture (REM, Agarwal et al. (2020)), its closest baselines, establishing iDQN as a relevant method to consider when aggregating significant advances to design a powerful value-based agent such as Rainbow (Hessel et al., 2018). We also perform further experimental studies to bring evidence of the intuition on which iDQN is built. We conclude the paper by discussing the limits of iDQN and pointing at some promising follow-up ideas.

## 2   Preliminaries

We consider discounted Markov decision processes (MDPs) defined as $\mathcal{M} = \langle \mathcal{S}, \mathcal{A}, \mathcal{P}, \mathcal{R}, \gamma \rangle$, where $\mathcal{S}$ is the state space, $\mathcal{A}$ is the action space, $\mathcal{P} : \mathcal{S} \times \mathcal{A} \times \mathcal{S} \to \mathbb{R}$ is the transition kernel of the dynamics of the system, $\mathcal{R} : \mathcal{S} \times \mathcal{A} \to \mathbb{R}$ is a reward function, and $\gamma \in [0, 1)$ is a discount factor (Puterman, 1990). A deterministic policy $\pi : \mathcal{S} \to \mathcal{A}$ is a function mapping a state to an action, inducing a value function $V^\pi(s) \triangleq \mathbb{E}\left[\sum_{t=0}^{+\infty} \gamma^t \mathcal{R}(S_t, \pi(S_t)) | S_0 = s\right]$ representing the expected cumulative discounted reward starting in state $s$ and following policy $\pi$ thereafter. Similarly, the action-value function $Q^\pi(s, a) \triangleq \mathbb{E}\left[\sum_{t=0}^{+\infty} \gamma^t \mathcal{R}(S_t, A_t) | S_0 = s, A_0 = a, A_t = \pi(S_t)\right]$ is the expected discounted cumulative reward executing action $a$ in state $s$, following policy $\pi$ thereafter. Q-learning aims to find a function $Q$ from which the greedy policy $\pi^Q(s) = \arg\max_a Q(\cdot, a)$ yields the optimal value function $V^*(\cdot) \triangleq \max_{\pi:\mathcal{S} \to \mathcal{A}} V^\pi(\cdot)$ (Puterman, 1990). The (optimal) Bellman operator $\Gamma^*$ is a fundamental tool in RL for obtaining optimal policies, and it is defined as:

$$(\Gamma^* Q)(s, a) \triangleq \mathcal{R}(s, a) + \gamma \int_{\mathcal{S}} \mathcal{P}(s, a, s') \max_{a' \in \mathcal{A}} Q(s', a') \mathrm{d}s', \tag{1}$$

for all $(s, a) \in \mathcal{S} \times \mathcal{A}$. It is well-known that Bellman operators are contraction mappings in $L_\infty$-norm, such that their iterative application leads to the fixed point $\Gamma^* Q^* = Q^*$ in the limit (Bertsekas, 2015). We consider using function approximation to represent value functions and denote $\Theta$ the space of their parameters. Thus, we define $\mathcal{Q}_\Theta = \{Q(\cdot|\theta) : \mathcal{S} \times \mathcal{A} \to \mathbb{R}|\theta \in \Theta\}$ as the set of value functions representable by parameters of $\Theta$.

## 3   Related Work

To provide an overview of the related work, we propose to view the related algorithms from the perspective of their behavior in the space of $Q$-functions, which we denote by $\mathcal{Q}$. Due to the curse of dimensionality, covering the whole space of $Q$-functions with function approximators is practically infeasible, as it requires a large number of parameters. Therefore, the space of *representable Q-functions* $\mathcal{Q}_\Theta$ only covers a small part of the whole space $\mathcal{Q}$. We illustrate this in Figure 1a by depicting the space of representable $Q$-functions as a subspace of $\mathcal{Q}$. One can deduce two properties from this gap in dimensionality. First, the optimal $Q$-function $Q^*$ is a priori not representable by any chosen function approximator. Second, the same is true for the optimal Bellman operator $\Gamma^*$ applied to a representable $Q$-function. That is why in Figure 1a, both functions $Q^*$ and $\Gamma^* Q$ are drawn outside of $\mathcal{Q}_\Theta$. Additionally, thanks to the contracting property of the optimal Bellman operator $||\Gamma^* Q - Q^*||_\infty \leq \gamma ||Q - Q^*||_\infty$, we know that the iterated $Q$ given by $\Gamma^* Q$ is at least $\gamma$ closer to the optimal $Q^*$ than the initial $Q$ (Bertsekas, 2015). The goal of most value-based methods is to learn a $Q$-function that is as close as possible to the projection of the optimal $Q$-function on the space of representable $Q$-functions, shown with a dotted line in Figure 1a.

This perspective allows us to represent a variety of Q-learning algorithms proposed so far in an intuitive way in a single picture. For example, Figure 1a depicts how Deep Q-Network (DQN) by Mnih et al. (2015) works. With a target network $\bar{Q}_0$, DQN aims at learning the iterated target network $\Gamma^* \bar{Q}_0$, also called "target", using an online network $Q_1$. The loss used during training is shown in red.

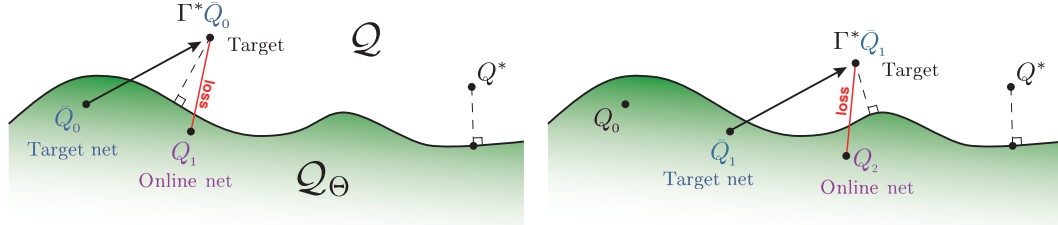

(a) DQN learns the optimal Bellman iteration of the $Q$-function represented by the target network $\bar{Q}_0$. It uses an online network $Q_1$ for that purpose.

(b) DQN updates the target network to make the target change. This means moving one Bellman iteration forward in the space of $Q$-functions.

Figure 1: Graphical representation of DQN in the space of $Q$-functions.

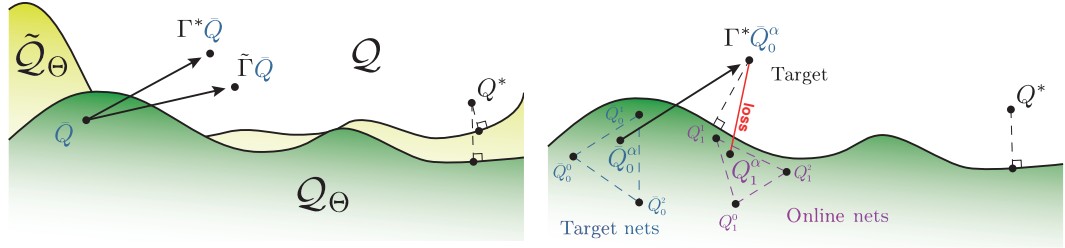

(a) Two common ways of improving DQN is to develop a more efficient empirical Bellman operator, noted $\tilde{\Gamma}\bar{Q}$, or to modify the space of representable $Q$-functions, noted $\tilde{\mathcal{Q}}_\Theta$.

(b) REM uses multiple $Q$-functions to explore areas in the space of $Q$-functions. A random convex combination of the stored $Q$-functions is sampled at each gradient step.

Figure 2: Graphical representations of DQN variants in the space of $Q$-functions.

In the optimal case, after a pre-defined number of training steps, the online network should represent the projection of the iterated target network onto the space of representable $Q$-functions (shown with a dotted line). This perspective also gives a way to understand the hyper-parameter related to the frequency at which the target network is updated. It is the number of training steps before moving to the next Bellman iteration. When the target network is updated, it will be equal to the online network, and the next Bellman iteration will be computed from there, as shown in Figure 1b. It is important to note that in DQN, the empirical Bellman operator is used instead of the optimal Bellman operator. The term included in the loss at every gradient step is a stochastic estimation of the optimal Bellman iteration.

## 3.1 DQN Variants

The DQN paper has inspired the community to develop further methods which improve its efficiency. A large number of those algorithms focuses on using a better empirical Bellman operator (Van Hasselt et al. (2016), Fellows et al. (2021), Sutton (1988)). For instance, double DQN (Van Hasselt et al., 2016) uses an empirical Bellman operator designed to avoid overestimating the return. This results in a different location of the Bellman iteration, as shown in Figure 2a. Other approaches consider changing the space of representable $Q$-functions (Wang et al. (2016), Osband et al. (2016)). The hope is that the projection of $Q^*$ on the space of representable $Q$-function is closer than for the classical neural network architecture chosen in DQN. It is important to note that adding a single neuron to one architecture layer can significantly change the space of representable $Q$-function. Wang et al. (2016) showed that performance can be increased by including inductive bias in the neural network architecture. This idea can be understood as a modification of the space of $Q$-functions, as shown in Figure 2a where the new space of representable $Q$-function is colored in yellow. Furthermore, algorithms such as Rainbow (Hessel et al., 2018) leverage both ideas. Other approaches, however, such as prioritized replay buffer (Schaul et al., 2015), cannot be represented in the picture.

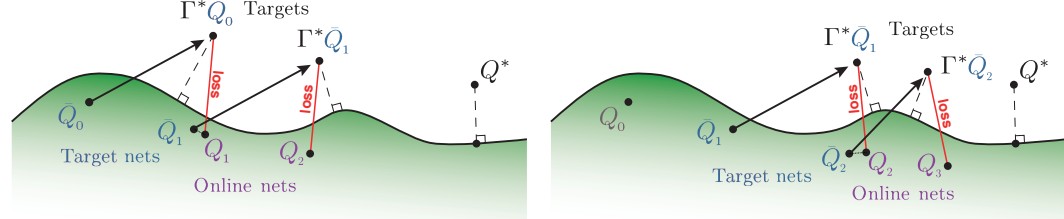

(a) iDQN learns two Bellman iterations at once. It uses a second target network $\bar{Q}_1$ to learn the second Bellman iteration. This second target network stays close to the first online network $Q_1$.

(b) iDQN updates the online and target networks to move one Bellman iteration ahead. Each network takes the value of the network corresponding to the following iteration.

Figure 3: Graphical representation of iDQN in the space of $Q$-functions.

## 3.2 Random Ensemble Mixture

Among the variants of DQN, ideas involving learning several $Q$-functions (Osband et al. (2016), Agarwal et al. (2020)) are particularly close to our method. Even if they are close, they remain orthogonal in the sense that they can be combined with our idea to create a more powerful agent. Random Ensemble Mixture (REM, Agarwal et al. (2020)) has been shown to be state-of-the-art for DQN variants with several $Q$-functions. Instead of exploring the space of $Q$-functions point by point as DQN does, REM moves in this space by exploring area by area, where the areas are the convex hull of the $Q$-functions stored in memory. As represented by the red line in Figure 2b, the loss used by REM is

$$\mathcal{L}(\theta) = \mathbb{E}_{(s,a,r,s')\sim\mathcal{D}}\left[\mathbb{E}_{\alpha\sim\Delta}[l(\delta^\alpha(s,a,r,s'|\theta))]\right],$$
$$\text{with } \delta^\alpha(s,a,r,s'|\theta) = Q_1^\alpha(s,a|\theta) - r - \gamma\max_{a'}\bar{Q}_0^\alpha(s',a'|\bar{\theta})$$

where $\theta$ denotes the parameters of the online network and $\bar{\theta}$ the target parameters, $\mathcal{D}$ is the replay buffer, $\Delta$ is the standard simplex and $l$ is the Huber loss (Huber, 1992), and $Q_i^\alpha = \sum_k \alpha_k Q_i^k, i \in \{0, 1\}$. For a Bellman iteration $i$, the $k^{\text{th}}$ learnt $Q$-function is noted $Q_i^k$. Figure 4a shows how this loss is computed with the neural network's architecture used in REM.

## 4 Iterated Deep Q-Networks

We propose an approach built on top of DQN. The main idea emerges naturally from the representation developed in Section 3. In DQN, Figure 1 illustrates that to learn two Bellman iterations, we first need to wait until the first iteration is learned, and then we need to update the target before starting to learn the second iteration. Conversely, we propose to use a second online network that learns the second Bellman iteration while the first Bellman iteration is being learned. The target for the second online network is created from a second target network that is frequently updated to be equal to the first online network. Figure 3a shows how iDQN behaves in the space of $Q$-function. It is important to understand that in iDQN, both online networks are learned at the same time. This idea can be further applied, as more $Q$-functions can be considered for learning the following Bellman iterations. Repeating the same schema, we can learn a following Bellman iteration by adding a new online network that would use a new target network set to be equal to the last online network. In Figure 3a, it would mean adding an online network $Q_3$, learnt to be the projection of the target $\Gamma^*\bar{Q}_2$, where $\bar{Q}_2$ is a target network periodically updated to the value of $Q_2$. This iterative process can be continued until memory usage becomes an issue. Once again, the loss can be observed on the representation in Figure 3a (see red lines), it writes

$$\mathcal{L}(s,a,r',s'|\theta) = \sum_{k=1}^{K}\left(Q_k(s,a|\theta) - r - \gamma\max_{a'}\bar{Q}_{k-1}(s',a'|\bar{\theta})\right)^2 \tag{2}$$

where $\theta$ is the online parameters and $\bar{\theta}$ the target parameters, $K$ is the number of Bellman iterations considered at once. The $k^{\text{th}}$ learned $Q$-function corresponds to the $k^{\text{th}}$ Bellman iteration and is noted $Q_k$. The way the loss is computed from the neural network's architecture is presented in Figure 4a.

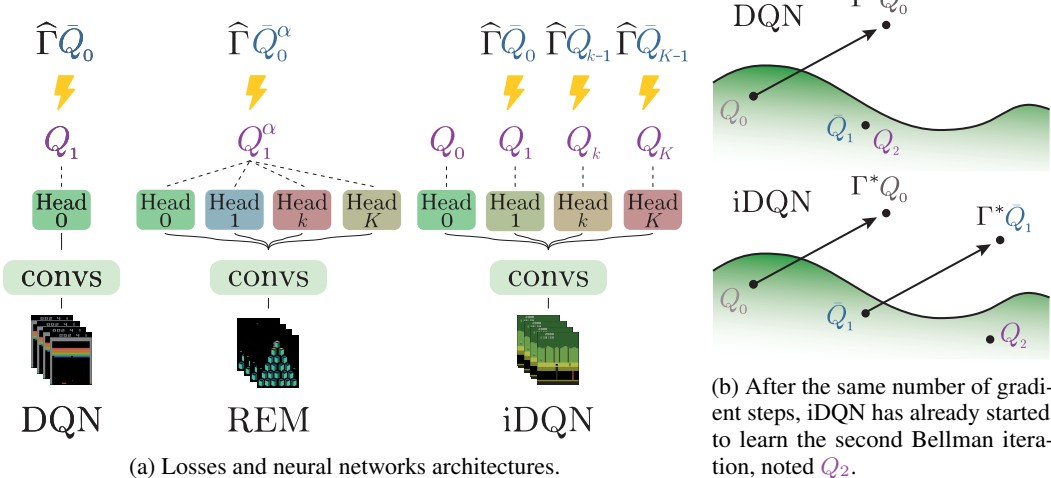

(a) Losses and neural networks architectures.

(b) After the same number of gradient steps, iDQN has already started to learn the second Bellman iteration, noted $Q_2$.

Figure 4: Understanding the loss of iDQN. In Figure 4a, the dotted lines link the outputs of the neural networks to the mathematical objects they represent. The flash signs stress how the information flows from the target(s) $\widehat{\Gamma}\bar{Q}$, considered fixed, to the online network(s) $Q$, on which the optimization is being done. For any $Q$-function $Q$, $\widehat{\Gamma}Q = \mathbb{E}_{s,a,s'}[\mathcal{R}(s,a) + \gamma \max_{a'} Q(s',a')]$ is the empirical Bellman operator.

In iDQN, updating the target networks does not bring the target parameters to the next Bellman iteration like in DQN. It simply refines their positions to be closer to the online networks to allow better estimates of the iterated $Q$-functions. To be able to go further in the Bellman iterations, we periodically consider a new online $Q$-function to learn and discard the first online and target network, as shown in Figure 3b. For this example, $Q_3$ is now considered and $Q_1$ and $\bar{Q}_0$ are left aside. We call this procedure a *rolling step*. In practice, the rolling step is simple to implement. A new head to the neural network of Figure 4a is added, with the index $K + 1$ and the first head is removed. It leads us to introduce a new hyper-parameter that indicates at which frequency the rolling step is performed. It is worth noticing that if $K$ is set to 1 and if the rolling step frequency is synchronized with the target update frequency in DQN, then we recover DQN, i.e., iDQN with K = 1 is equal to DQN.

In DQN, the actions are drawn from the online network. For iDQN, one needs to choose from which of the multiple online networks to sample. One could stick to DQN and choose the first online network. One could also use the last online network since it is supposed to be the one that is closer to the optimal $Q$-function, or one could pick an online neural network at random as it is done in Bootstrapped DQN (Osband et al., 2016). We do not consider taking the mean as REM proposes because the online $Q$-functions are expected to follow a specific arrangement in space. Taking their mean could lead to unwanted behavior. We investigate these sampling strategies in Section 5.1. One can see how iDQN uses the same algorithm as DQN through its pseudo-code shown in Algorithm 1 in the supplementary material. The only two changing parts are the behavioral policy and the loss.

### 4.1 Understanding the Loss of iDQN

The crucial advantage of iDQN is coming from the loss. In addition to the loss used in DQN, it contains $K - 1$ more terms. Those terms concern the future Bellman iterations, hence the fact that *iDQN allows for a better learning of the Bellman iterations*. In practice, each Bellman iteration is learned with $K$ times more gradient steps than in DQN while having the same overall number of gradient steps. This means that each selected sample is used $K$ times more or, in other words, that each network sees $K$ times more samples. As mentioned earlier, updating the target in DQN moves the learning procedure one Bellman iteration further. The same goes for the rolling step for iDQN. From the fact that each Bellman iteration is learned with $K$ times more gradient steps than in DQN, we can allow iDQN to perform the rolling step more frequently than DQN updates the target, which means that iDQN will do more Bellman iterations at the end of the training, while still learning each iteration better than DQN. Figure 4b pinpoints the advantage of iDQN with $K = 2$ over DQN. There we assume that the update target frequency of DQN is equal to the rolling step frequency in

iDQN. When DQN updates its target network $Q_0$ to $\bar{Q}_1$, the new online network $Q_2$ is located at $\bar{Q}_1$. When the rolling step is performed for iDQN, $\bar{Q}_1$ is located at the same position as the $\bar{Q}_1$ of DQN[1] but $Q_2$ has already been learnt and is closer to the optimal $Q$-function than the $Q_2$ of DQN. This phenomenon is even stronger when $K$ increases. Another way to understand iDQN is to see iDQN as a way to pre-train the next online $Q$-functions instead of taking them equal to the online network as it is done in DQN.

# 5  Experiments

We evaluate our proposed algorithm on 54 Atari 2600 Games (Bellemare et al., 2013)[2]. Many implementations of Atari environments along with classical baselines are available online (Castro et al. (2018), D'Eramo et al. (2021), Raffin et al. (2021), Huang et al. (2022)). We choose to mimic the implementation choices made in Dopamine (Castro et al., 2018) since it is the only one to release the evaluation metric for all relevant baselines to our work and the only one to use the evaluation metric recommended by Machado et al. (2018). Namely, we use the *game over* signal to terminate an episode instead of the *life* signal. The input given to the neural network is a concatenation of 4 frames in gray scale of dimension 84 by 84. To get a new frame, we sample 4 frames from the Gym environment (Brockman et al., 2016) configured with no frame skip, and we apply a max pooling operation on the 2 last gray scale frames. We use sticky actions to make the environment stochastic (with $p = 0.25$). The training performance is the one obtained during learning. By choosing an identical setting as Castro et al. (2018) does, we can take the baselines' training performance of Dopamine without the need to train them again ourselves. To certify that the comparison is fair, we compared our version of DQN to their version and concluded positively (see Figure A of the supplementary material).

**Hyperparameter tuning.**    The hyperparameters shared with DQN are kept untouched. The two additional hyperparameters (rolling step frequency and target parameters update frequency) were set to follow our intuition on their impact on the performance. As a reminder, the frequency at which the rolling step is performed is comparable to the target update frequency in DQN. Since iDQN allows more gradient steps per iteration, we set this hyperparameter to be 25% lower than the target update frequency in DQN (6000 compared to 8000). To further ensure that our code is trustworthy, Figure A in the supplementary material shows that DQN achieves similar training performances than iDQN with $K = 1$ and the rolling step frequency is set to be equal to the target update frequency of DQN. It is important to note that decreasing the target parameters update results in a more stable training but also a higher delay with the online networks which can harm the overall performance. We set it to 30, allowing 200 target updates per rolling step. We choose $K = 5$. This choice is further discussed in Section 5.1. To make the experiments run faster, we designed the $Q$-functions to share the convolutional layers. Additionally, we consider the first layers of the neural network useful for extracting a feature representation of the state space. This is why this choice can potentially be beneficial to our algorithm. Further details about the hyperparameters can be found in the supplementary material.

**Performance metric.**    As recommended by Agarwal et al. (2021), we choose the interquartile mean (IQM) of the human normalized score to report the results of our experiments with shaded regions showing pointwise 95% percentile stratified bootstrap confidence intervals. IQM is a trade-off between the mean and the median where the tail of the score distribution is removed on both sides to consider only 50% of the runs. 5 seeds are used for each game.

**Main result.**    iDQN greatly outperforms DQN (Adam) on the aggregation metric, proposed in Agarwal et al. (2021). Figure 5a shows the IQM human normalized score over 54 Atari games according to the number of frames sampled during the training. In the last millions of frames, iDQN reaches a higher IQM human normalized score than DQN (Adam). iDQN performs better than REM as well, showing that our approach should be preferred when using multi-head $Q$ networks. We do not consider other variants of DQN to be relevant baselines to compare with. The ideas used in Rainbow, IQN or Munchausen DQN (Vieillard et al., 2020) can be included in iDQN algorithm to build an

---

[1]The loss in iDQN is additive and includes the DQN loss. Thus both $Q_1$ are located at the same position.

[2]We excluded the game *Atlantis* due to the significantly higher training time.

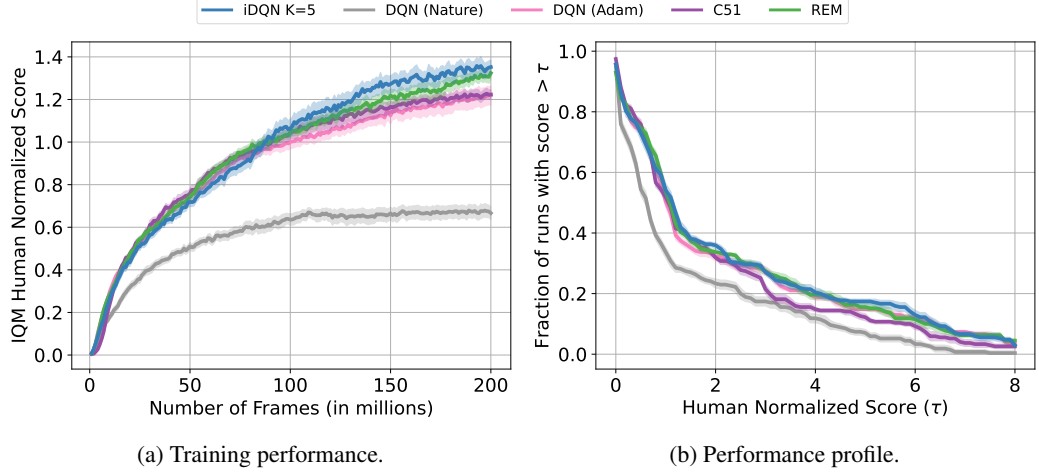

(a) Training performance.       (b) Performance profile.

Figure 5: iDQN outperforms DQN (Nature), DQN (Adam), C51 and REM. DQN (Nature) uses the RMSProp optimizer (Tieleman et al., 2012) while DQN (Adam) uses Adam (Kingma & Ba, 2015).

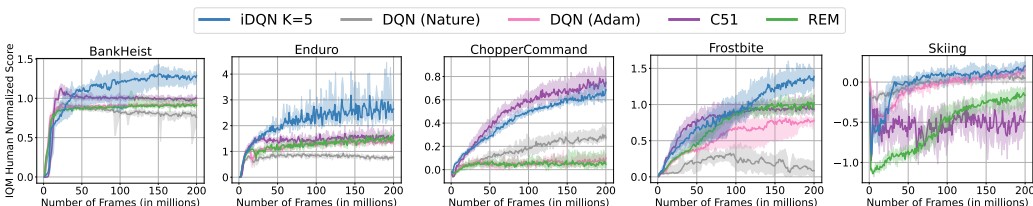

Figure 6: 5 Atari games where different behaviors can be observed.

even more powerful agent. We further discuss it in Section 5.1. To visualize the distribution of final scores, we plot the performance profile in Figure 5b. It shows the fraction of runs with a higher final score than a certain threshold given by the $X$ axis. In some ranges of human normalized score, iDQN statistically dominates DQN. For example, there are more games in which iDQN achieves $1.5$ times a human performance ($\tau = 1.5$) than DQN. It is harder to distinguish iDQN and REM because the gap in final performance between iDQN and REM is smaller than between iDQN and DQN. In Figure 6, we selected 5 games where different behaviors can be observed. On some games like *BankHeist* and *Enduro*, iDQN overtakes all its baselines. Interestingly, in *ChopperCommand*, DQN (Adam) and REM fail at outperforming DQN (Nature), while iDQN is comparable with C51 in performance. This shows that efficiently learning the Bellman iterations plays an important role in some environments. In *Frostbite*, REM is more efficient than DQN (Adam). iDQN outperforms both algorithms in this game, being the only one achieving superhuman performances. Finally, in *Skiing*, REM and C51 failed to be better than a random agent, while iDQN sticks to the performance of DQN (Adam). We believe this behavior comes from the fact that iDQN is close to DQN in principle, which minimizes the risk of failing when DQN succeeds. We present the training performance of all the remaining games in Figure B of the supplementary material.

## 5.1 Ablation Studies

We perform several ablation studies to showcase the different behaviors of iDQN. We first investigate the importance of the number of Bellman iterations $K$ taken into account in the loss. As shown in Figure 7 for the games *Asteroids* and *Asterix*, increasing $K$ to 10 iterations could be beneficial. In *Qbert*, the gain seems not certain. We believe further tuning of hyperparameters should bring iDQN with $K = 10$ to yield better performances than iDQN with $K = 5$. We insist that no hyperparameter tuning has been performed to generate this plot. In order to have the same number of gradient steps per Bellman iteration for $K = 5$ and $K = 10$, we simply halved the frequency at which the rolling step is performed for $K = 10$, bringing it to 3000 since we doubled $K$. Interestingly, the performance

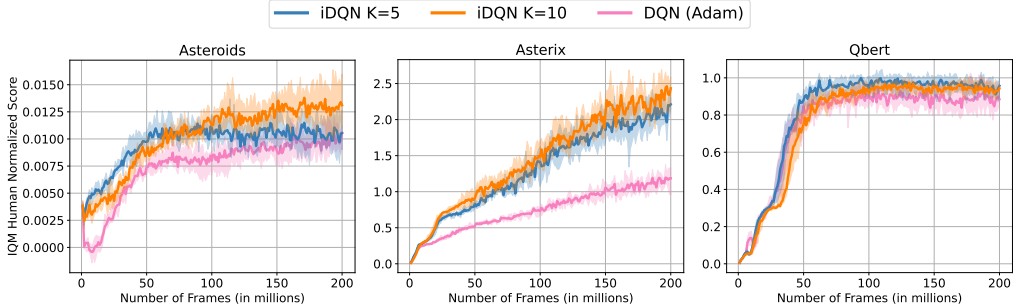

Figure 7: Ablation study on the number of Bellman iterations $K$ taken into account in the loss.

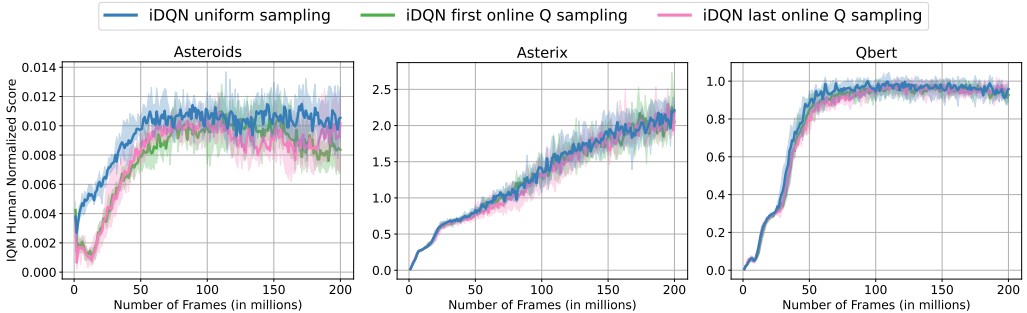

Figure 8: Ablation study on the way actions are sampled to interact with the environment. Actions can be sampled from an online $Q$-function taken at random (in blue), from the first online $Q$-function (in green), or from the last $Q$-function (in pink).

never drops in the 3 considered games. Therefore, *we recommend increasing $K$ as much as the computational resources allow it.*

In Section 4, we mentioned several sampling strategies. Figure 8 illustrates the different possibilities mentioned earlier: sampling from a uniform online $Q$-function, sampling from the first online $Q$-function like DQN does, or sampling from the last online $Q$-function, it is supposed to be the closest to the optimal $Q$-function. No significant difference exists except for the game *Asteroids*, where sampling from a uniform online $Q$-function seems to yield better performance throughout the training. We believe that the increase in performance comes from the fact that the online Q-functions generate a wider variety of samples compared to only sampling from the first or last online Q-function. *We recommend sampling actions from an online Q-function chosen at random.*

iDQN heavily relies on the fact that the learned Q functions are located at different areas in the space of $Q$-functions. We computed the standard deviation of the output of the learned $Q$-functions during the training in Figure 9 to verify this assumption. The figure shows that the standard deviation among the $Q$-function is indeed greater than zero across the 3 studied games. Furthermore, we can observe that the standard deviation decreases during training, hinting that they become increasingly closer. This matches the intuition that at the end of the training, the $Q$-functions should be close to the projection of the optimal $Q$-function, hence being close to each other.

## 6 Discussion

In Figure 10, we compare iDQN with other powerful baselines to show the gap between those baselines and our approach, which does not use the benefit of a prioritized replay buffer and a $n$-step return. The curves for other algorithms shown in Figure 10 depict the publicly available metrics for those algorithms.[3] The training performance of IQN and Munchausen DQN without the $n$-step return would be interesting to analyze, but our limited resources do not allow us to run those baselines. The major improvement of Rainbow over C51 is made by using a prioritized replay buffer and

---

[3]Often different metrics are used for different algorithms, making the comparison not straightforward.

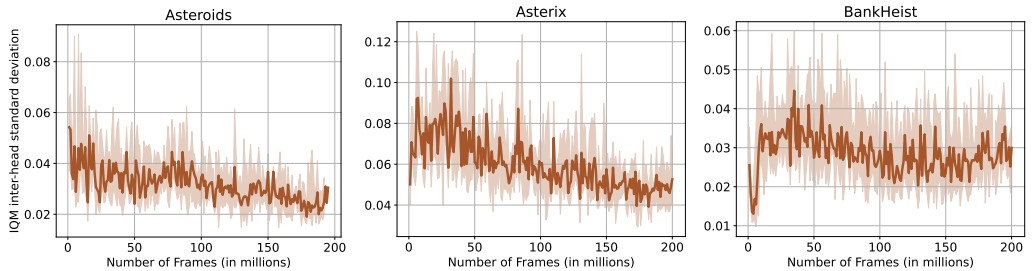

Figure 9: Standard deviation of the online networks' output averaged over 3200 samples at each iteration.

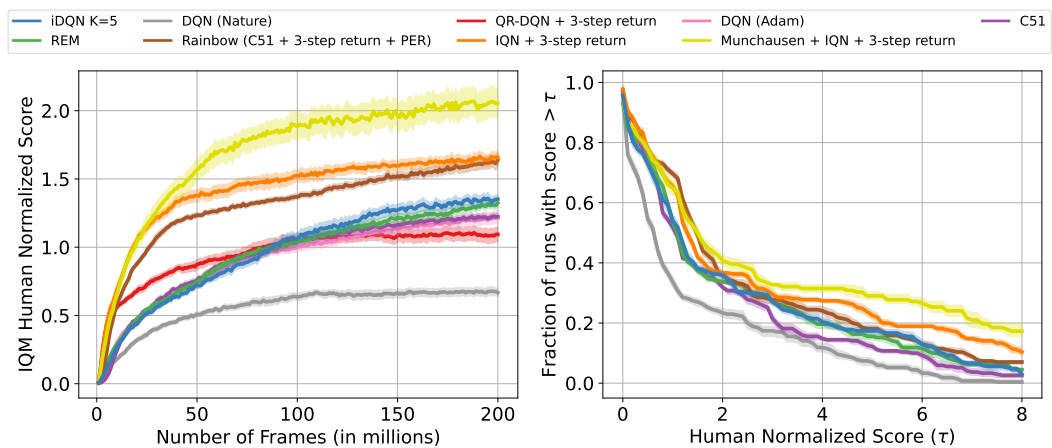

Figure 10: Training performance (left) and performance profile (right) of iDQN and other orthogonal methods. QR-DQN (Dabney et al., 2018) and DQN (Adam) have been removed from the performance profile for clarity.

adding a 3-step return which gives hope for following works to study the potential strength of any Rainbow-like iDQN approach. The training performances of iDQN on 54 Atari games along with those more advanced methods are available in Figure C of the supplementary materials.

Our approach introduces two new hyperparameters, rolling step frequency and target parameters update frequency, that need to be tuned. However, we provide a thorough understanding of their effects to mitigate this drawback. While extreme values of some hyperparameters were found to be harmful to the performance of DQN, e.g., changing the target update frequency, little variation of the presented values was found to have only a small impact on the overall performance. Regarding the resources needed to train an iDQN agent, more computations are required to get the gradient of the loss compared to DQN. Thanks to the ability of JAX (Bradbury et al., 2018) to parallelize the computation, iDQN with $K = 5$ only requires 1 to 2 times more time to run. With the released code base, each run presented in this paper can be run under 3 days on an NVIDIA RTX 3090.

## 7 Conclusion

In this paper, we have presented a way to learn the Bellman iterations more efficiently than DQN. The underlying idea of iDQN comes from an intuitive understanding of DQN's behavior in the space of $Q$-functions. It allows each $Q$ estimate to be learned with more gradient steps without increasing the overall number of gradient steps. iDQN outperforms its closest baselines, DQN and REM, on the Atari 2600 benchmark. While we proposed an approach to $Q$-learning that focuses on the projection step of the Bellman iterations, an interesting direction for future work would be to investigate which other past improvements of DQN, in combination with iDQN, would lead to a new state-of-the-art for value-based methods.

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
