# OpenReview forum: "Iterated Deep Q-Network: Efficient Learning of Bellman Iterations for Deep Reinforcement Learning"
_NeurIPS.cc/2023/Conference — Submitted to NeurIPS 2023_

### Official Review · Reviewer_UiiX · 2023-07-03

**Soundness:** 3 good
**Presentation:** 3 good
**Contribution:** 3 good
**Rating:** 7
**Confidence:** 4

**Summary:**

This paper focuses on learning the projection of the empirical Bellman operator's iteration on the space of function approximator (neural model). This being done through increasing the number of gradient steps using multiple heads with a certain form of update. While retaining the same total number of gradient steps and samples compared to common approaches, the proposed method seems to provide better results (at the cost of retaining multiple heads and more computation).

**Strengths:**

The idea is interesting, novel and practical. The paper also **experimentally** shows noticeable improvement over various baselines.

**Weaknesses:**

- The presented method is quite simple and could have been presented much more efficiently with simple math and direct explanation rather than lengthy discussions over multiple figures.

- The choice of $K$ seems to have a significant impact on the behaviour, which also varies depending on the domain. Suggestion: some formal analysis (e.g., the algorithm's variance) could be useful to provide better insight about what to expect from larger $K$ in terms of other characteristics such as the transition kernel.

**Questions:**

- The main idea of the paper is simple: using multiple heads for a Q network and compute the Bellman loss for each of them using its preceding head’s target, which provides multiple grad steps at once for a given transition (also performs the rolling step more frequently). This has been presented through a long (and figure-based) discussion. While I did like the figure-based discussion of the “related work” section, I believe a set of equations can be far more efficient to explain the method and compare it against other techniques. Honestly, looking at Eq 2 with a short paragraph of explanation is sufficient (and way better) to understand what is going on. The current form is ponderous and hard to follow.

- In section "Preliminaries": the policy is assumed to be **deterministic**. Why? (this is a significant limitation and calls for more discussion.)


- It’s been said both in the abstract and section 4.1 that iDQN concern the future Bellman iterations. Looking at the loss function (Eq 2), all the $Q_k$’s corresponding loss terms are computed using the same transition tuple. I do not see why it has anything to do with “future Bellman iterations”.


- I liked figures 1 & 2 with the corresponding discussion of section 3. One missing point here is the value-mapping idea: to functionally map $Q$ into a difference space and perform the learning, then map back to the space of $Q$. This basically alters the actual $Q$ space and is inherently different from choosing a different approximator. Hence, the approximator’s space boundary may retain better properties such as remaining homogeneously close to the $Q^*$ for a larger subset of states. See this paper [and possibly others]: https://arxiv.org/abs/2203.07171  note that this paper also discusses orchestration of multiple mappings...

- For the ensemble techniques, this paper is of particular interest, which also provides a neat way of diversification (hence, requiring smaller ensemble size): https://arxiv.org/abs/2110.01548.


Other comments:

- L78--79 --> “... it is at least $\gamma$ closer ...” --> if being picky, this is not accurate. Rather, the distance is shrunk by $\gamma$ (as a multiplier, not a subtractor).

- Beginning of L86 -->. “In the optimal case...” --> what do you mean?

- Eq 2 --> what is $r’$ ?

**Limitations:**

While the authors mentioned at the end of Introduction that "We conclude the paper by discussing the limits of iDQN ...", they apparently forgot to do so! No discussion of limitations is provided.

---

> ### Author Rebuttal · Authors · 2023-08-09
>
> We thank the reviewer for the extensive feedback. It seems that this work has raised questions we are happy to discuss.
>
> **Weaknesses**
>
> > 2. The choice of $K$ [...]
>
> Several reviewers have raised this point. We kindly ask the review to refer to point $I$ of the general answer that addresses this specific point.
>
> **Questions**
>
> > 1. The main idea of the paper is simple: [...]
>
> We thank the reviewer for pointing this out. We understand that the reviewer suggests us to clarify Section $4$. This is why we propose a new version in which the core idea is explained before discussing the benefit of iDQN. We believe the following can increase the understandability of Section $4$:
>
> We propose an approach built on top of DQN. In practice, this new idea consists in changing the loss of DQN such that it includes $K$ temporal difference errors instead of one:
>
> $\mathcal{L}(s, a, r, s' \vert \theta) = \sum_{k=1}^K \left(Q_k(s, a \vert \theta) - r - \gamma \max_{a'} \bar{Q}_{k-1}(s', a' \vert \bar{\theta}) \right)^2$
>
> where $\theta$ is the online parameters and $\bar{\theta}$ the target parameters. The $k^{th}$ learned $Q$-function corresponds to the $k^{th}$ Bellman iteration and is denoted $Q_k$. The way the loss is computed from the neural network’s architecture is presented in Figure $4a$. One can see how the $Q$-functions are chained one after the other to learn the Bellman iterations.
>
> In iDQN, updating the target networks does not bring the target parameters to the next Bellman iteration like in DQN. It simply refines their positions to be closer to the online networks to allow better estimates of the iterated $Q$-functions. To go further in the Bellman iterations, we periodically consider a new online $Q$-function and discard the first target network in the sense that the index $k$ in the loss would now go from $2$ to $K+1$. We call this procedure a rolling step. In practice, the rolling step is simple to implement. A new head to the neural network of Figure $4a$ is added, with the index $K + 1$, and the first head is removed. It leads us to introduce a new hyper-parameter that indicates at which frequency the rolling step is performed. It is worth noticing that if K is set to $1$ and if the rolling step frequency is synchronized with the target update frequency in DQN, then we recover DQN, i.e., iDQN with $K = 1$ is equal to DQN.
>
> This main idea emerges naturally from the representation developed in Section $3$. In DQN, Figure $1$ illustrates that to learn $2$ Bellman iterations, we first need to wait until the first iteration is learned, and then we need to update the target before starting to learn the second iteration. Conversely, we propose to use a second online network that learns the second Bellman iteration while the first Bellman iteration is being learned. The target for the second online network is created from a second target network that is frequently updated to be equal to the first online network. Figure 3a shows how iDQN behaves in the space of $Q$-function. It is important to understand that in iDQN with $K = 2$, both online networks are learned at the same time. As explained earlier in this section, we can learn a following Bellman iteration by adding a new online network $Q_3$ that would use a new target network $\bar{Q}_2$ set to be equal to the last online network $Q_2$ as shown in Figure 3b. In the meantime, the target and online network are discarded to keep memory usage constant. In practice, the choice of $K$ can be increased until memory usage becomes an issue.
>
> In DQN, the actions are ...[Continuing with Section $4$ as it is in the submission.]
>
> > 2. In section "Preliminaries": [...]
>
> The policy can be stochastic. We will write "A policy" and remove the term "deterministic".
>
> > 3. It’s been said both in the abstract and [...]
>
> All $Q_k$'s in the loss are trained with the same samples. This is a key advantage of the proposed method (iDQN): each $Q$ function is trained with more samples, or putting it differently, the samples are used more often. It is important to point out that the samples are not related to the Bellman iterations. They are generated by the behavioral policy (see Section 5.1, line 251). The Bellman iterations are learned by a pair of target $Q$ functions and an online $Q$ function whose index indicates the Bellman iteration count. In DQN, only one Bellman iteration is trained at a time through the online $Q$ function. However, iDQN does the same and learns the $K - 1$ following Bellman iterations through the $K - 1$ online $Q$ functions added 'after' the first online network. The pairs of target and online $Q$ functions are attached together to form a chain, as it is shown in Figure 3a.
>
> > 4. I liked figures 1 & 2 [...]
>
> We thank the reviewer for pointing us to this interesting work. We will add it to the related work section.
>
> > 5. For the ensemble techniques, [...]
>
> We thank the reviewer for broadening our scope of related works. We will happily add it to the related work section.
>
> Other comments:
>
> > L78--79 --> [...]
>
> We agree with the reviewer the proposed version is more accurate. We will add it to the final version of the paper.
>
> > Beginning of L86 -->. “In the optimal case...” --> what do you mean?
>
> For each Bellman iteration, the goal is to train the online network to be as close as possible to the target computed from the target network. The equality is unlikely because the target can be located outside of the space of representable functions (the space of representable $Q$ functions is shown in green in all figures). This is why 'in the optimal case', the online network is located at the projection of the target on the space of representable $Q$ functions.
>
> > Eq 2 --> what is $r'$?
>
> This is a mistake. Thank you for pointing it out. It should be $r$. We will add it to the final version of the paper.

---

> > ### Comment · Reviewer_UiiX · 2023-08-13
> >
> > I appreciate the authors' feedback and serious consideration. I am willing to increase the score.
> >
> > > choice of $K$ + newly added theory:
> >
> > Thanks for the new plots. Together with the formal exposition, it's a significant improvement. A couple of points:
> >
> > * I noticed $k$ goes from 0 to $K$, meaning $K+1$ in total; I think this is a typo. In the theorem, it's from 1 to $K$ though.
> >
> > * In my view, this sentence: "DQN minimizes only one term of the bound, while iDQN minimizes the whole sum of terms we have influence over" is actually what summarizes iDQN the best.
> >
> > * I cannot wrap my head around Figure E (in the submitted PDF). The x-axis is $k$, which should have gone from 1 to $K$ for each colour, right? Say orange should go from 1 to 10, but it looks like all colours go all the way to 20! Can you clarify?
> >
> > > The main idea of the paper is simple...
> >
> > Suggestion (minor): The sentence is a bit confusing. To avoid confusion with n-step learning, perhaps say it like this: "... a particular ensemble of K one-step temporal difference instead of just one".

---

> > > ### Author Response · Authors · 2023-08-13
> > >
> > > We thank the reviewer for the valuable suggestions. We are glad to see that our rebuttal has been appreciated.
> > >
> > > > 1. I noticed $k$ goes from $0$ to $K$, meaning $K + 1$ in total; I think this is a typo. In the theorem, it's from $1$ to $K$ though.
> > >
> > > It is actually wanted. To learn $K$ Bellman iterations, we need $K + 1$ $Q$-functions. There are indeed $K$ terms in the loss but the first term uses $Q_0$ ($Q_{k - 1}$ with $k = 1$).
> > >
> > > > 3. I cannot wrap my head around Figure $E$ (in the submitted PDF). The x-axis is $k$, which should have gone from $1$ to $K$ for each colour, right? Say orange should go from $1$ to $10$, but it looks like all colours go all the way to $20$! Can you clarify?
> > >
> > > The Reviewer is correct. We simply repeat the process for every experiment until we reach $20$ Bellman iterations so that each experiment has the chance to perform the same amount of Bellman iterations. More precisely, after a pre-defined number of gradient steps, we only keep the last $Q$-function and learn the $K$ following Bellman iterations by duplicating the kept $Q$-function $K$ times and by using the loss of iDQN.
> > >
> > > > 4. Suggestion (minor): The sentence is a bit confusing. To avoid confusion with n-step learning, perhaps say it like this: "... a particular ensemble of K one-step temporal difference instead of just one".
> > >
> > > Thank you for the suggestion. We agree that this new version is more effective:
> > >
> > > We propose an approach built on top of DQN. In practice, this new idea consists in changing the loss of DQN such that it is composed of a particular ensemble of $K$ one-step temporal difference instead of one: ...
> > >
> > > We are glad that the Reviewer is willing to increase the score. However, we see that the score is still the original one. Please let us know if more clarifications are needed for increasing the score.

---

### Official Review · Reviewer_PHSC · 2023-07-06

**Soundness:** 3 good
**Presentation:** 2 fair
**Contribution:** 3 good
**Rating:** 6
**Confidence:** 3

**Summary:**

This paper presents Iterated Deep Q-Network (iDQN), a new DQN-based algorithm that incorporates multiple Bellman iterations into the training loss. The paper highlights the limitations of traditional RL methods that only consider a single Bellman iteration and proposes iDQN as a solution to improve learning. The algorithm leverages the online network of DQN to build targets for successive online networks, taking into account future Bellman iterations. The paper evaluates iDQN against relevant baselines on 54 Atari 2600 games and demonstrates its benefits in terms of approximation error and performance.

**Strengths:**

1. The proposed iDQN algorithm introduces a novel approach to incorporate multiple Bellman iterations into the training loss, addressing the limitations of traditional RL methods.
2. The paper provides a well-structured review of relevant literature, discussing the behavior of various Q-learning algorithms in the space of Q-functions. This analysis helps in understanding the motivation behind iDQN and its relationship with other methods.
3) The empirical evaluation on selected Atari games demonstrates the superiority of iDQN over its closest baselines, DQN and Random Ensemble Mixture. This provides empirical evidence of the effectiveness of the proposed approach.

**Weaknesses:**

1. It would be helpful if the paper included more comparisons with widely-known baselines in the field. While the paper compares iDQN to DQN and Random Ensemble Mixture, it would be valuable to see how iDQN performs against other popular RL algorithms like R2D2.
2. Some parts of the paper could be further clarified to improve the reader's understanding. For example, the explanation of the loss function and the graphical representations of DQN variants could be made more concise and intuitive.

**Questions:**

1. Can you demonstrate the performance table of Atari 26/54 games?
2. Can you continue to improve the demonstration of Figure 1,2,3? It's difficult to understand what they are demonstrating.
3. Will you provide code or implementation details?

**Limitations:**

It would be helpful if the paper included more comparisons with widely-known baselines in the field. This paper has no negative social impacts.

---

> ### Author Rebuttal · Authors · 2023-08-09
>
> We thank the reviewer for the valuable feedback.
>
> **Weaknesses**
>
> > 1. It would be helpful if the paper included more comparisons with widely-known baselines in the field. While the paper compares iDQN to DQN and Random Ensemble Mixture, it would be valuable to see how iDQN performs against other popular RL algorithms like R2D2.
>
> Several reviewers have raised this point. We kindly ask the review to refer to point $II$ of the general comment, which tackles this specific point.
>
> **Questions**
>
> > 1. Can you demonstrate the performance table of Atari 26/54 games?
>
> In the supplementary materials, we provide the learning curves of iDQN along with other methods on all games. We chose to represent the performance profiles to show the distribution of the final scores as recommended by [1]. If the reviewers think it adds clarity, we can gladly add the table of the final scores of iDQN and other relevant DQN methods in addition to the learning curves and the performance profiles.
>
> > 2. Can you continue to improve the demonstration of Figure 1,2,3? It's difficult to understand what they are demonstrating.
>
> We are highly interested in increasing the clarity of the figures. We believe the understandability of the figures can be improved from their caption. Please find updated versions of the figure captions that we would like to add in the final version of the paper:
>
> Figure 1: Graphical representation of DQN in the space of $Q$-functions $\mathcal{Q}$. The space of parameterized $Q$-functions $\mathcal{Q}\_{\Theta}$ is shown in green. The optimal $Q$-function $Q^*$ is in most cases not finitely representable, hence not belonging to $\mathcal{Q}\_{\Theta}$. DQN makes use of a target network $\bar{Q}\_{k-1}$ to learn its optimal Bellman iteration $\Gamma^* \bar{Q}\_{k-1}$, also called target, with an online network $Q_k$. Each iteration is learned by minimizing the distance between the target $\Gamma^* \bar{Q}_{k-1}$ and the online network $Q_k$ (see the red line).
>
> (a) Starting from a random $Q$-function $\overline{Q}_0$, the first Bellman iteration is learned via an online network $Q_1$ using stochastic gradient descent.
>
> (b) After a predefined number of gradient steps, the online network is frozen and is used as a target network, noted $\overline{Q_1}$, to learn the second Bellman iteration. $Q_2$ is the online network learning the second Bellman iteration.
>
> Figure 2: Graphical representation of DQN variants in the space of $Q$-functions $\mathcal{Q}$.
>
> (a) One common way of improving DQN is to develop a more efficient empirical Bellman operator, denoted $\tilde{\Gamma}$. This can lead to more stable behavior, hence easing the learning process. Another way is to modify the space of representable Q-functions, denoted $\tilde{\mathcal{Q}}_{\Theta}$, so that the optimal $Q$-function $Q^*$ is closer to this space, increasing the possibilities of learning a $Q$-function being closer to $Q^*$.
>
> (b) To better explore the space of representable $Q$-functions $\mathcal{Q}_{\Theta}$, REM keeps in memory $3$ target networks $\overline{Q}_0^0$, $\overline{Q}_0^1$ and $\overline{Q}_0^2$ and $3$ online networks $Q_1^0$, $Q_1^1$ and $Q_1^2$. Similarly to DQN, at each gradient step, REM uses a target $Q$-function $\overline{Q}_0^{\alpha}$ computed as a random convex combination of the stored target networks. This target $Q$-function learns its optimal Bellman iteration $\Gamma^*\overline{Q}_0^{\alpha}$ with an online network $Q_1^{\alpha}$ computed from the same convex combination of the stored online networks.
>
> Figure 3: Graphical representation of iDQN in the space of $Q$-functions denoted $\mathcal{Q}$. iDQN makes use of a target networks $\overline{Q}$ to learn their optimal Bellman iterations $\Gamma^*\overline{Q}$, also called target, with an online network $Q$. Each iteration is learned by minimizing the distance between the target $\Gamma^*\overline{Q}_{k-1}$ and the online network $Q_k$ (see the red lines). The update target frequency regulates the distance between the target network and the online network corresponding to the same Bellman iteration (shown in dotted points). In this Figure, iDQN learns $2$ Bellman iterations at once.
>
> (a) In this example, iDQN starts by learning the $2$ first Bellman iterations. This Figure depicts what happens after all the networks have been initialized randomly and after a few gradient steps and updates of the target parameters have been done.
>
> (b) A rolling step is performed to learn the third Bellman iteration. This is been done by incrementing all the indexes by one unit. As a result, the first target network $\overline{Q_0}$ is discarded, and a new online network $Q_3$ is learned.
>
> > 3. Will you provide code or implementation details?
>
> Please find the code and the implementation details in the supplementary materials. We plan to publish the code via a GitHub link upon acceptance.
>
> [1] Agarwal, R., Schwarzer, M., Castro, P. S., Courville, A. C., and Bellemare, M. Deep reinforcement learning at the edge of the statistical precipice. Advances in Neural Information Processing Systems, 34, 2021.

---

> > ### Comment · Area_Chair_EYn4 · 2023-08-20
> > **RE:**
> >
> > The author's rebuttal appears to address your concerns. Can you clarify if you have additional questions/concerns?

---

### Official Review · Reviewer_G4ak · 2023-07-09

**Soundness:** 3 good
**Presentation:** 3 good
**Contribution:** 2 fair
**Rating:** 4
**Confidence:** 3

**Summary:**

The paper considers the problem of how to get accurate approximations of optimal Q-functions. The paper introduces a new algorithm called iterated DQN (iDQN). iDQN incorporates multiple consecutive Bellman iterations into the training process, which aims to allow for better approximation of optimal action-value functions. It uses the online network of DQN to build a target for a second online network, and so on, for considering future Bellman iterations. The authors conducted several experiments based on Atari games by comaping iDQN with baseline methods, including DQN and Random Ensemble Mixture.

**Strengths:**

- Significance and Originality: The topic that the paper studies - how to learn the Bellman iterations efficiently, is an important topic in reinforcement learning. The authors propose a simple yet effective method, called iterated DQN, to improve the learning efficiency. Specifically, iterated DQN uses a second online Q-network for learning the second Bellman iteration simultaneously, where the target for the second online Q-network is created from a second target network according to the first online network. In this way, the loss can include K-1 more terms compared to the original loss used in DQN. The way for using such kind of ensemble seems novel, which allows for improved efficiency.

- Clarity: The paper is well-written and easy to follow, with very clear illustrations for the update for DQN, REM, and iterated DQN as in Figures 1-4.

**Weaknesses:**

- Quality: The paper presents a simple yet effective idea, but it could be further strengthened particularly in theoretical and empirical analysis. First, the authors could provide a theoretical guarantee for iterated DQN by examining its convergence speed, in addition to the intuitive explanation given in Section 4.1. This would lend credibility to their claims. Second, the empirical validation raises concerns, as iterated DQN's performance is only marginally better than that of previous baseline methods such as DQN (Adam), C51, and REM. This modest improvement does not strongly support the paper's assertions. Lastly, it would be beneficial for the authors to include a memory comparison, as employing more Q-networks may lead to increased memory costs, which is an important consideration for practical applications.

**Questions:**

1. Can the authors also develop a theoretical guarantee for iterated DQN by analyzing its convergence speed besides the intuitive explanation in Section 4.1? It would be better if  the claim is also demonstrated theoretically.

2. My main concern for the paper is about the empirical validation part. Iterated DQN actually performs very closely to previous baseline methods including DQN (Adam), C51, and REM, with quite small improvement margin, which does not support the claim in the paper well.

3. Can the authors also show the memory comparison, as using more Q-networks can induce a larger memory cost?

**Limitations:**

The authors have discussed some of the limitations by considering other value-based methods.

---

> ### Author Rebuttal · Authors · 2023-08-09
>
> We thank the reviewer for the feedback and insightful questions.
>
> > 1. Can the authors also develop a theoretical guarantee for iterated DQN by analyzing its convergence speed besides the intuitive explanation in Section 4.1? It would be better if the claim is also demonstrated theoretically.
>
> Several reviewers have raised this point. Please refer to point $I$ of the general comment that tackles this point.
>
> > 2. My main concern for the paper is about the empirical validation part. Iterated DQN actually performs very closely to previous baseline methods including DQN (Adam), C51, and REM, with quite small improvement margin, which does not support the claim in the paper well.
>
> Several reviewers have raised this point. Please refer to point $II$ of the general comment that tackles this point.
>
> > 3. Can the authors also show the memory comparison, as using more Q-networks can induce a larger memory cost?
>
> Suppose we note $C$, the memory necessary to store the parameters for the convolutional layers, and $F$, the memory used for the parameters of the fully connected layers. DQN needs $2 (C + F)$, the $2$ comes from the fact that there is a target and an online network. For iDQN, the memory used is $2 (2C + (K + 1)F)$. There is a target and an online network as well. This is why there is a $2$ on the left. Then, as shown in the supplementary material in Figure D, $2$ sets of convolutional parameters are stored along with $K + 1$ heads. More precisely, the classical architecture used in Atari games requires $16$ MB of memory while iDQN with $K = 5$ requires $92$ MB and $168$ MB for $K = 10$. It is worth noticing that those quantities are negligible compared to the space the replay buffer needs. It can reach several GBs even with some memory optimization tricks. We would be glad to add this analysis to the final version.

---

> > ### Comment · Area_Chair_EYn4 · 2023-08-20
> > **RE:**
> >
> > Did the authors point II address your primary concern about the empirical validation?

---

> > > ### Comment · Reviewer_G4ak · 2023-08-21
> > >
> > > I would like to thank the authors for the response and the AC for the kind reminder. Upon reviewing the supplemental PDF, I would like to express my concern about the clarity of Figure F. The figure appears to present conflicting information, which raises doubts about the claim made in the paper regarding the superiority of iIQN over IQN in the just two considered Atari games.
> > >
> > > In StarGunner, the performance of iIQN is actually observed to be less efficient than that of IQN. Furthermore, the performance of both iIQN and IQN seems to converge towards the end of the run, with marginal differences in their outcomes. Consequently, based on these observations, I am inclined to believe that while the paper presents a well-motivated and intriguing approach, the experimental results do not provide robust justification for the claim put forth. Therefore, I keep the rating for now.

---

> > > > ### Author Response · Authors · 2023-08-21
> > > > **Clarifications about Figure F**
> > > >
> > > > We thank the reviewer for considering our responses. We are happy to clarify the purpose of the additional experiment which seems to be the only remaining concern of the review.
> > > >
> > > > > The figure appears to present conflicting information, which raises doubts about the claim made in the paper regarding the superiority of iIQN over IQN in the just two considered Atari games.
> > > >
> > > > The purpose of Figure $F$ is not to demonstrate that iIQN is superior to IQN since we would need to evaluate both algorithms on all Atari games. With this figure, we want to show that iDQN can be combined with other variants of DQN and improve their performance. We think that this is clearly shown by Figure $F$ for the game Frostbite.
> > > >
> > > > > In StarGunner, the performance of iIQN is actually observed to be less efficient than that of IQN. Furthermore, the performance of both iIQN and IQN seems to converge towards the end of the run, with marginal differences in their outcomes.
> > > >
> > > > We agree with the reviewer that in StarGunner, IQN yields higher return at the beginning of the training. However, for the sake of clarity, we did not mention that for iDQN + IQN (iIQN), we use $N = 32$ quantiles and $N' = 32$ target quantiles while IQN uses $N = 64$ quantiles and $N' = 64$ target quantiles. This choice has been made to be able to finalize the experiments on time for the deadline of the rebuttal. This difference has been shown to be important by the authors of IQN in the earlier stage of the training. In [1], they write: "As expected, we found that $N$ has a dramatic effect on early performance [...]. Additionally, we observed that $N'$ affected performance very differently than expected: it had a strong effect on early performance [...].". Concerning the final performance of iIQN at the end of the training on StarGunner, not only the interquartile mean (IQM) is higher than IQN and IQN + $3$-step return but it is higher than all the other variants of DQN. For this reason, we found this result interesting.
> > > >
> > > > We invite the reviewer to have a look at the results shared with Reviewer VNTR where we show evidence of the fact that iDQN can yield better performance over another variant of DQN (DQN + $n$-step return) on different Atari games than the one used for the analysis with IQN.
> > > >
> > > > [1] Dabney, Will, et al. "Implicit quantile networks for distributional reinforcement learning." International conference on machine learning. PMLR, 2018.

---

### Official Review · Reviewer_A74j · 2023-07-12

**Soundness:** 3 good
**Presentation:** 4 excellent
**Contribution:** 3 good
**Rating:** 7
**Confidence:** 4

**Summary:**

This work proposes an extension to DQN aimed at improving projection steps in Q value updates. There are two main contributions of the paper:

- The paper intuitively explains the Q-value learning characteristics of DQN variants caused by a mismatch between the optimal Bellman operator and the set of representable Q functions.
- The authors propose the iterated DQN (iDQN) method, which keeps track of a collection of K online Q-functions. When updating these Q networks, the previous Q function is used as the target network.

Experiment results show that iDQN outperforms a collection of DQN variants on the standard Atari benchmark. Further ablation studies explore the effect of K and sampling strategies for iDQN and conclude that bigger K and uniform sampling of Q networks are in general preferable.

**Strengths:**

- The figures explaining the projection characteristics of existing DQN variants are very intuitive and provide excellent motivation for the proposed method.
- The iDQN method, the newly introduced hyperparameters, and the experiment settings are communicated clearly and transparently.
- The results on Atari are convincing.
- The ablation studies on the choice of K and sampling method are insightful.

**Weaknesses:**

- The figures used to explain the projection behaviors of DQN variants are created for illustration, but not from actual experiments.
- As discussed by the authors, iDQN doesn't outperform more recent DQN variants which employ other tricks to improve performance.

**Questions:**

- The online Q networks are known to be unstable during training. Could later Q networks suffer from compounding stability issues as they use previous online Q networks as targets?
- Connecting back to weakness #1, is it possible to create a visualization of the projection behaviors in some toy environments?

**Limitations:**

The authors discussed how iDQN is not able to outperform more recent DQN variants using other tricks. It would also be nice if the authors can discuss the training stability of iDQN.

---

> ### Author Rebuttal · Authors · 2023-08-09
>
> We thank the reviewer for the useful suggestions and comments.
>
> **Weaknesses**
>
> > 2. As discussed by the authors, iDQN doesn't outperform more recent DQN variants which employ other tricks to improve performance.
>
> Several reviewers have raised this point. We kindly ask the review to refer to point $II$ of the general answer that addresses this specific point.
>
> **Questions**
> > 1. The online Q networks are known to be unstable during training. Could later Q networks suffer from compounding stability issues as they use previous online Q networks as targets?
>
> Thank you for pointing this out. This is precisely why we use a target network so that the targets are not updated as frequently as the online networks. This is enough to avoid instability in practice. More precisely, we update the targets every $30$ training steps.
>
>
> > 2. Connecting back to weakness #1, is it possible to create a visualization of the projection behaviors in some toy environments?
>
> We understand the point raised by the reviewer. In the additional PDF file, we added an additional experiment on a toy offline problem: Linear Quadratic Regulator (see Figure G). In this problem, the state and action spaces are continuous and one-dimensional. The dynamics are linear: for a state $s$ and an action $a$, the next state is given by $s' =  0.8 s - 0.9 a$, and the reward is quadratic $r(s, a) = 0.5 s^2 + 0.4 sa -0.5 a^2$. We choose to parametrize the space of $Q$-function with $2$ parameters $(M, G)$ such that, for a state $s$ and an action $a$, $Q(s, a) = M a^2 + G s^2$. To avoid having a too large space of representable $Q$-functions, we constrain the parameter $M$ to be negative and parameter $G$ to be between $-0.4$ and $0.4$. Starting from some initial parameters, we performed $30$ gradient steps with a learning rate of $0.05$ using the loss of DQN and iDQN. Both figures show the space of representable $Q$-function $\mathcal{Q}_{\Theta}$ in green, the optimal $Q$-function $Q^*$, the initial $Q$-function $Q_0$ and its optimal Bellman iteration $\Gamma^* Q_0$. The projection of the optimal Bellman iteration is also shown with a dotted line. The figure on the left shows where the online network $Q_1$, computed from DQN, is after $30$ gradient steps. The figure on the right shows where the online networks $Q_1$ and $Q_2$ of iDQN are after $30$ gradient steps. As we claim in the paper, iDQN finds a $Q$-function $Q_2$ closer to the optimal $Q$-function $Q^*$ than $Q_1$ found by DQN. The figure on the left closely resembles Figure 1a. Likewise, the figure on the right looks like Figure 3a, showing that the high-level ideas presented in the paper are actually happening in practice.

---

> > ### Comment · Reviewer_A74j · 2023-08-20
> >
> > Thanks for adding the additional experiment! The toy example makes sense and supports the motivation of the paper.

---

> > > ### Author Response · Authors · 2023-08-21
> > >
> > > We are glad to see that the concerns raised by the reviewer have been cleared by our responses and additional experiments.

---

### Official Review · Reviewer_VNTR · 2023-07-23

**Soundness:** 2 fair
**Presentation:** 3 good
**Contribution:** 2 fair
**Rating:** 6
**Confidence:** 4

**Summary:**

In this paper, a new variant of DQN algorithm, iDQN, is proposed by replacing the classical Bellman iteration with several consecutive Bellman iterations and using multiple Q networks.
Intuitively, this new Bellman operator propogates reward sigals more efficiently thus speeds up learning, with the cost of more computation and memory.
As the number of consecutive Bellman iterations increases, it is shown that the learning performance of iDQN in Atari games is also increased.

**Strengths:**

As far as I know, the proposed method is novel. The new algorithm, together with several baselines, are tested in 54 Atari games.
Many illustrative pictures are included to help understand the new Bellman operator.
The paper is generally well-written.

**Weaknesses:**

1. It will make this work much better if a theoretical analysis of the proposed Bellman operator is provided, such as convergence speed, the affect of the number of Q networks, etc.
2. The performance of iDQN is only slightly better than baselines, such as DQN(Adam). A summarized result (e.g. the first figure in [DQN Zoo](https://github.com/deepmind/dqn_zoo)) can make the comparison in Atari games much clearer.
3. Missing related works about ensemble methods + DQN, e.g. Averaged DQN and Maxmin DQN.
4. Misssing baselines: DQN + n-step return. Both iDQN and this baseline try to accelerate the propogations of reward signals. Furthermore, although it is mentioned that "We do not consider other variants of DQN to be relevant baselines to compare with.", more explanations are necessary.
5. It is claimed that "Our approach introduces two new hyperparameters, rolling step frequency and target parameters update frequency, that need to be tuned. However, we provide a thorough understanding of their effects to mitigate this drawback. ". However, I don't find the understanding thorough enough.

**Questions:**

See Weaknesses.

**Limitations:**

The limitations of iDQN are that it takes more memory and computation than DQN. Jax is used to parallelize the computation, making the training time increase acceptable.

---

> ### Author Rebuttal · Authors · 2023-08-09
>
> We thank the reviewer for the insightful comments and feedback.
>
> > 1. It will make this work much better if a theoretical analysis of the proposed Bellman operator is provided, such as convergence speed, the affect of the number of Q networks, etc.
>
> Several reviewers have raised this point. Therefore, we provided a general answer in point $I$ of our overall rebuttal response. Please refer to that response.
>
> > 2. The performance of iDQN is only slightly better than baselines, such as DQN(Adam). A summarized result (e.g. the first figure in DQN Zoo) can make the comparison in Atari games much clearer.
>
> It seems that there has been a misunderstanding. Figure 5a shows the same plot as in DQN Zoo with the Interquartile Mean (IQM) instead of the median as recommended by [1]. As described in [1], the IQM removes the worst 25% and the best 25% of the scores before taking the mean to avoid the influence of outlayers while keeping more than one point. Figure 10a shows the same plot with more variants of DQN. It is worth noting that all DQN variants can be combined with our approach (iDQN). Therefore, outperforming DQN is the key step. In future work, we plan to evaluate other improvements on top of iDQN, but experiments on Atari take some time.
>
> > 3. Missing related works about ensemble methods + DQN, e.g. Averaged DQN and Maxmin DQN.
>
> Thank you for pointing this out. In case of acceptance, we will add those works in the related works section.
>
> > 4. Missing baselines: DQN + n-step return. Both iDQN and this baseline try to accelerate the propogations of reward signals. Furthermore, although it is mentioned that "We do not consider other variants of DQN to be relevant baselines to compare with.", more explanations are necessary.
>
> Several reviewers have raised this point. Please refer to point $II$ of the general answer that addresses this specific point.
>
> > 5. It is claimed that "Our approach introduces two new hyperparameters, rolling step frequency and target parameters update frequency, that need to be tuned. However, we provide a thorough understanding of their effects to mitigate this drawback. ". However, I don't find the understanding thorough enough.
>
> We thank the reviewer for raising this point. We truly believe that we can provide some intuition on how to set these hyperparameters. First, the rolling step frequency defines the speed at which we learn the Bellman iterations. Problems in which the environment is highly stochastic will require more gradient steps to learn a Bellman iteration hence the need to increase the rolling step frequency. Conversely, problems with sparse rewards and long horizons will be faster to learn with a high rolling step frequency because more Bellman iterations are needed to reach good performance. Second, the target update frequency indicates the speed at which the target networks follow the online networks. Once again, highly stochastic problems will benefit from having a small target update frequency since the positions of the online networks are more likely to be noisy. Conversely, problems with sparse rewards and long horizons can benefit from having the target networks closely following the online networks.
> We will add this clarification in the final version of the paper.
>
> [1] Agarwal, R., Schwarzer, M., Castro, P. S., Courville, A. C., and Bellemare, M. Deep reinforcement learning at the edge of the statistical precipice. Advances in Neural Information Processing Systems, 34, 2021.

---

> > ### Comment · Reviewer_VNTR · 2023-08-12
> > **Further comment**
> >
> > Thank you for your response.
> >
> > I'm willing to increase the score from 4 to 5 since theoretical analysis is provided. However, I still have a concern about the effectiveness of the proposed method compared to DQN + n-step return, given that the proposed method uses more computation and memory resources.

---

> > > ### Author Response · Authors · 2023-08-13
> > > **Clarification on the link between iDQN and $n$-step return**
> > >
> > > We thank the reviewer for considering our answer and increasing the score.
> > >
> > > We understand the concern of the reviewer. At first sight, it seems that $n$-step return and iDQN yield similar benefits but it is not the case. iDQN is a method that allows for better learning of each Bellman iteration without increasing the total number of gradient steps. This advantage comes from the loss that takes into account several Bellman iterations at each gradient step. However, $n$-step return provides a different way of estimating the return. iDQN + $n$-step return is practically feasible and does not incur more memory or computational time over iDQN just like DQN + $n$-step return does compared to DQN. As explained in point $II$ of the general answer, the reason why we did not run iDQN + $n$-step return is simply because of the high computational cost coming from Atari games. Therefore, we do not consider $n$-step return to be a baseline but a possible addition to iDQN left for future work.
> > >
> > > This clarification would fit perfectly in Section 3.1 where we introduce other empirical Bellman operators. Line $98$, we cite Sutton (1988) [1] in which $n$-step return is introduced. We would gladly add this clarification to this paragraph in the revised version of the paper.
> > >
> > > [1] Sutton, R. S. Learning to predict by the methods of temporal differences. Machine Learning, 3(1): 9–44, August 1988. URL http://www.cs.ualberta.ca/~sutton/papers/sutton-88.pdf.

---

> > > > ### Comment · Reviewer_VNTR · 2023-08-14
> > > > **Response to clarification**
> > > >
> > > > Thank you for your clarification.
> > > >
> > > > I understand the difference between iDQN and DQN+n-step return. However, I still think it would be better if there is a comparison between iDQN and DQN+n-step return. Compared to DQN+n-step return, iDQN uses more computation and memory due to the usage of n-heads, while DQN+n-step return is a simpler solution. You do not have to test them in all Atari games. Using a subset of Atari games or gym tasks should be ok.

---

> > > > > ### Author Response · Authors · 2023-08-21
> > > > >
> > > > > We thank the reviewer for replying to our comment.
> > > > >
> > > > > Using Dopamine's code base, we ran DQN + $3$-step return on $2$ Atari games (Asterix and DemonAttack) over $5$ seeds for $100M$ frames. Due to the lack of time, we did not run DQN + $3$-step return for $200M$ frames. We report the interquartile mean (IQM) every 10 epoch here in comparison with iDQN, $K = 5$:
> > > > >
> > > > > | Asterix         |        10        |        20        |       30        |        40        |        50        |        60        |    70       |      80       |     90      |     100      |
> > > > > |-----------          |:----------------:|:----------------:|:---------------:|:----------------:|:----------------:|:----------------:|:----------------:|:---------:|:---------:|-----------:|
> > > > > | DQN + 3-step return |0.30 (+- 0.02)|  0.36   (+- 0.03)|  0.40   (+- 0.02)|  0.44   (+- 0.04)|  0.44   (+- 0.04)|  0.46   (+- 0.06)|  0.49   (+- 0.03)|  0.51   (+- 0.05)|  0.51   (+- 0.06)|  0.56   (+- 0.06)|
> > > > > |    iDQN K = 5       |0.30 (+- 0.01)|__0.54__ (+- 0.07)|__0.69__ (+- 0.03)|__0.72__ (+- 0.06)|__0.80__ (+- 0.08)|__0.94__ (+- 0.04)|__1.05__ (+- 0.12)|__0.98__ (+- 0.10)|__1.21__ (+- 0.17)|__1.35__ (+- 0.21)
> > > > >
> > > > > | DemonAttack         |        10        |        20        |       30        |        40        |        50        |        60        |    70       |      80       |     90      |     100      |
> > > > > |-----------          |:----------------:|:----------------:|:---------------:|:----------------:|:----------------:|:----------------:|:----------------:|:---------:|:---------:|-----------:|
> > > > > | DQN + 3-step return |   1.09  (+- 0.06)|   1.26  (+- 0.12)|   1.47  (+- 0.09)|  1.58   (+- 0.16)|  1.76   (+- 0.17)|  2.09   (+- 0.18)|  2.14   (+- 0.14)|2.60 (+- 0.53)|2.90 (+- 0.26)|  3.04   (+- 0.25)|
> > > > > |    iDQN K = 5       |__1.86__ (+- 0.25)|__3.22__ (+- 0.29)|__3.20__ (+- 0.28)|__3.53__ (+- 0.31)|__3.17__ (+- 0.37)|__2.95__ (+- 0.63)|__3.09__ (+- 0.43)|3.23 (+- 0.32)|3.25 (+- 0.43)|__3.61__ (+- 0.23)|
> > > > >
> > > > > One can see that iDQN outperforms DQN + 3-step return on these two games. Our point here is not to show that iDQN statistically dominates DQN + 3-step return on all games (this would be too computationally expensive) but to show that iDQN brings superior results in this randomly selected subset of Atari games.

---

> > > > > > ### Comment · Reviewer_VNTR · 2023-08-22
> > > > > >
> > > > > > Thank you for the response. I have increased the score to 6.

---

> > > > > > > ### Author Response · Authors · 2023-08-22
> > > > > > >
> > > > > > > We thank the reviewer for considering our answer and increasing the score.

---

### Author Rebuttal · Authors · 2023-08-09

**General comment to all reviewers.**

We thank all the reviewers for their valuable feedback. For each reviewer, we address their concern with specific answers. Here, we summarize the common points, describe the content of the rebuttal PDF in the attachment, and provide answers that we will add to the final version of the paper.

### $I.$ Theoretical analysis of iDQN has been requested by several reviewers (asked by reviewers VNTR, G4ak, and UiiX).

We thank the reviewers for pointing out the value of providing a theoretical justification for our proposed iDQN algorithm. Indeed, although we focus mainly on empirical evaluation, it is possible to make a statement about why iDQN should, in principle, be expected to improve upon the performance of DQN. Namely, we can invoke Theorem 3.4 from [1] on error propagation for Approximate Value Iteration (AVI):

**Theorem 3.4 [1].** Let $K \in \mathbb{N}^*$, $\rho$, $\nu$ two distribution probabilities over $\mathcal{S} \times \mathcal{A}$. For any sequence $(Q_k)\_{k=0}^K \subset B \left(\mathcal{S} \times \mathcal{A}, R_{\gamma} \right)$ where $R_{\gamma}$ depends on reward function and discount factor, we have

$|| Q^* - Q^{\pi_K} ||\_{1, \rho} \leq C_{K, \gamma, R_{\gamma}} + \inf_{r \in [0, 1]} F(r; K, \rho, \gamma) \left(\sum_{k=1}^{K} \alpha_k^{2r} || \Gamma^*Q_{k - 1} - Q_k ||_{2, \nu}^{2} \right)^{\frac{1}{2}}$

where $\alpha_k$ and $C_{K, \gamma, R_{\gamma}}$ do not depend on the sequence $(Q_k)_{k=0}^K$. Function $F(r; K, \rho, \gamma)$ depends on the concentrability coefficients of the greedy policies w.r.t. the value functions.

In simpler words, this theorem bounds the approximation error at each iteration by a term that includes the sum of approximation errors until the current timestep, i.e., $\sum_{k=1}^{K}\alpha_k^{2r} || \Gamma^*Q_{k - 1} - Q_k ||_{2, \nu}^{2}$.

It can be seen that at iteration $k$, the DQN loss $( r + \gamma \max_{a'}Q_{k-1}(s', a') - Q_k(s, a) )^2$ is an unbiased estimator of the approximation error $|| \Gamma^* Q_{k - 1} - Q_k ||$. Likewise, the iDQN loss $\sum_{k=1}^K (r + \gamma \max_{a'}Q_{k-1}(s', a') - Q_k(s, a) )^2$ is an unbiased estimator of the sum of approximation errors $\sum_{k=1}^K || \Gamma^*Q_{k - 1} - Q_k ||$. From there, one can see that at each gradient step, DQN minimizes only one term of the bound, while iDQN minimizes the whole sum of terms we have influence over. Hence, at each gradient step, iDQN can lower the approximation error bound more than DQN.

We would be glad to clarify this further if needed and to include it in the final version of the paper.

We complement this theoretical analysis with an empirical evaluation of a low-dimensional offline problem Car-On-Hill [2], where the agent needs to drive an underpowered car to the top of a hill. It has a continuous state space and two possible actions: moving left or right. In this problem, the optimal value function $V^*$ can be computed via brute force [2]. Figure E in the rebuttal PDF shows the distance between the optimal value function $V^*$ and $V^{\pi_i}$, i.e., the value function of the greedy policy of the current action-value function estimate obtained with iDQN. This distance is plotted according to the Bellman iterations computed during the training for several values of $K$. We recall that iDQN with $K = 1$ is equivalent to DQN or, more precisely, FQI since it is an offline problem. The plot clearly shows that for higher values of $K$, iDQN performs better because it reaches lower approximation errors earlier during the training. This relates to the theorem previously described. By increasing the value of $K$, we increase the number of Bellman iterations taken into account for each gradient step. Hence we decrease the upper bound on the distance between the optimal value function and the current estimate, which is what is happening in the plot.

### $II.$ iDQN is not outperforming all considered variants of DQN (asked by reviewers VNTR, A74j, G4ak, and PHSC).

In this work, we focused on validating that iDQN outperforms DQN. Since iDQN is orthogonal to all other proposed improvements of DQN, one can expect that adding them to iDQN would further improve the results. Due to the high computational cost of running Atari games, we could not do a thorough analysis of these combinations yet. In our opinion, this would be the content of future work, similar to what Hessel et al. did in [3]. However, in the rebuttal PDF file, we provide some results that combine iDQN with $K=3$ and Implicit Quantile Networks (IQN) (see Figure F). In the two considered Atari games, iIQN (i.e., iDQN + IQN) outperforms iDQN and IQN, showing that not only is it technically feasible to combine those two algorithms but that it can yield better performance. Interestingly, iIQN even outperforms IQN + 3-step return. We would gladly include these results in the paper and release the code of iIQN upon acceptance.

### $III.$ Figures 1, 2, and 3 are illustrations not made from experimental results (asked by reviewer A74j).

This point only concerns one reviewer. Therefore, we only write the answer to reviewer A74j. Nonetheless, we invite reviewers interested in this point to read the answer.

[1] Farahmand, A. M. (2011). Regularization in Reinforcement Learning (Doctoral dissertation, University of Alberta).

[2] Ernst, D., Geurts, P., and Wehenkel, L. Tree-based batch mode reinforcement learning. JMLR, 6:503–556, dec 2005. ISSN 1532-4435.

[3] Hessel, M., Modayil, J., Van Hasselt, H., Schaul, T., Ostrovski, G., Dabney, W., Horgan, D., Piot, B., Azar, M., and Silver, D. Rainbow: Combining improvements in deep reinforcement learning. In Proceedings of the AAAI conference on artificial intelligence, volume 32, 2018.

---

### Decision · Program_Chairs · 2023-09-21

**Decision:**

Reject

**Comment:**

Initially, many of the reviewers found that the paper had limited theoretical justification. The authors provided a theoretical justification in the rebuttal, however, the theoretical setup is quite a bit removed from the practical algorithm and their claimed benefit is not precise enough to be a theoretical statement.

Reviewer's asked for additional comparisons with other approach and the authors conducted additional experiments. While the additional experiments were appreciated, some reviewers felt, that even after the rebuttal the experiments "do not provide sufficient support for the claims made".

Given the weak theoretical support, that the empirical gains are small, and the comparisons with other approaches is limited, I'm recommending rejection at this time. I encourage the authors to conduct the additional experiments as they mentioned they would do and resubmit.